# Phylogenetic inference of changes in amino acid propensities with single-position resolution

**Galya V. Klink**[1], **Olga V. Kalinina**[2], **Georgii A. Bazykin**[1,3]*

**1** Institute for Information Transmission Problems (Kharkevich Institute) of the Russian Academy of Sciences, Moscow, Russia, **2** Helmholtz Institute for Pharmaceutical Research Saarland (HIPS), Helmholtz Centre for Infection Research (HZI), Saarbrücken, Germany; Medical Faculty, Saarland University, Homburg, Germany, **3** Skolkovo Institute of Science and Technology, Skolkovo, Russia

* g.bazykin@skoltech.ru

**Data Availability Statement:** All relevant data are within the manuscript and its Supporting information files and on GitHub: https://github.com/GalkaKlink/d-test.

## Abstract

Fitness conferred by the same allele may differ between genotypes and environments, and these differences shape variation and evolution. Changes in amino acid propensities at protein sites over the course of evolution have been inferred from sequence alignments statistically, but the existing methods are data-intensive and aggregate multiple sites. Here, we develop an approach to detect individual amino acids that confer different fitness in different groups of species from combined sequence and phylogenetic data. Using the fact that the probability of a substitution to an amino acid depends on its fitness, our method looks for amino acids such that substitutions to them occur more frequently in one group of lineages than in another. We validate our method using simulated evolution of a protein site under different scenarios and show that it has high specificity for a wide range of assumptions regarding the underlying changes in selection, while its sensitivity differs between scenarios. We apply our method to the *env* gene of two HIV-1 subtypes, A and B, and to the HA gene of two influenza A subtypes, H1 and H3, and show that the inferred fitness changes are consistent with the fitness differences observed in deep mutational scanning experiments. We find that changes in relative fitness of different amino acid variants within a site do not always trigger episodes of positive selection and therefore may not result in an overall increase in the frequency of substitutions, but can still be detected from changes in relative frequencies of different substitutions.

## Author summary

Which amino acids are acceptable at a certain protein site can change with time. In viruses, for example, this can be due to changes in mechanisms of drug resistance and immune escape in the course of evolution. Here, we develop a method for detecting such changes from how evolutionary events are distributed over an evolutionary tree. Informally, we infer that a certain amino acid is favored in a certain group of lineages if substitutions giving rise to it repeatedly occur in the evolution of this group, and disfavored if

**Funding:** This work was supported by the Russian Foundation for Basic Research grant no. 18-34-00358 (recipient: G. V. K.). The funders had no role in study design, data collection and analysis, decision to publish, or preparation of the manuscript.

**Competing interests:** The authors have declared that no competing interests exist.

such substitutions are rare. In surface proteins of HIV-1 and influenza A, we find that changes in preferences detected with d-test match those observed in deep mutational scanning experiments. Our purely bioinformatic approach allows inference of changes in selection between lineages from sequences alone, shedding light on the functional differences between strains or species even in the absence of any structural or functional data.

## Introduction

A particular amino acid at a specific position of a protein may confer different fitness in different species. Such changes in amino acid propensities can occur due to substitutions at other epistatically interacting sites in the course of evolution. Indeed, accounting for co-occurrence of amino acids improves predictions of the rates and patterns of substitutions [1] and polymorphism [2,3], as well as the pathogenic potential of mutations [4,5] and their functional effects in experiments [6–9]. Changes in amino acid propensities can also happen due to shifts in protein fitness landscape caused by a changing environment, e.g. shifts in immune pressure on proteins of pathogens [10–13].

Variability of amino acid propensities has been inferred from statistical analyses of pooled data from numerous amino acid positions of large protein alignments [14,15]. In particular, it has been shown to bias the phylogenetic distribution of homoplasies–repeated substitutions giving rise to the same allele: parallelisms, convergences [16–20] and reversals [21,22]. This is because the fixation probability of a mutation, and therefore the rate of a substitution, giving rise to an allele increases with its conferred fitness [23]. Therefore, substitutions producing the same allele are expected to be more frequent in groups of species where this allele confers high fitness [24]. On a phylogeny, this can be observed as clustering of homoplasies in these groups of species, and such clustering has been interpreted as evidence for changes in fitness landscapes. Most existing methods [16–22] aggregate data across multiple sites to increase power, and therefore do not allow to track changes in amino acid propensities at individual sites.

Here, we develop an approach, referred to as the d-test, for identification of amino acid variants at a protein site that confer different fitness in two predefined taxa (such as species or viral strains). This approach uses the phylogenetic distribution of substitutions at this site, and is based on the logic that substitutions that occur in lineages closely related to one of the taxa will be probably more advantageous in it. Using simulations, we show that our method performs well on phylogenies that include hundreds of samples or more. By applying the d-test to large viral phylogenies, we infer sites in the gp160 protein (encoded by the *env* gene) of HIV-1 that have undergone a change in the relative fitness of different amino acids between two strains representative of two highly divergent HIV-1 subtypes; and sites in the hemagglutinin (HA) protein of influenza A that have undergone such a change between two strains representative of two influenza subtypes. We then validate these findings using the fitness differences measured for these strains directly in deep mutational scanning (DMS) experiments [25–27].

## Results

### Inferring variable fitness amino acids

Using the reconstructed phylogenetic positions of substitutions, we analyze how homoplasic (parallel or convergent, i.e., giving rise to the same amino acid) and divergent (giving rise to different amino acids) substitutions at the same amino acid site are distributed relative to each other over the phylogeny. We propose that an excess of homoplasic substitutions in a set of

related lineages implies that the descendant allele confers higher fitness in these lineages, compared to the fitness conferred by it in more distantly related lineages. By contrast, a deficit of substitutions into an allele implies that it confers lower fitness at this group of lineages [28,29].

Specifically, we aim to identify individual amino acids at specific sites that arise at significantly different rates between the two parts of the phylogenetic tree. Each part is defined as the set of branches that are phylogenetically close to a predefined node (i.e., belong to the "phylogenetic neighborhoods" of this node); this node can be external (corresponding to observed species or strains) or internal (corresponding to reconstructed ancestors). For this, we start by considering the first of the two nodes as the "focal" node. For each amino acid at each site, we then test two hypotheses: that the phylogenetic positions of substitutions into this amino acid are biased towards ("proximal amino acids") or away from ("distal amino acids") the focal node (Fig 1). Next, we consider the second species as the focal node, and repeat the procedure. Finally, we assume that the fitness conferred by an amino acid has changed between the two focal nodes if it was classified as proximal to one of them and as distal from the other, and designate such amino acids as variable fitness amino acids.

## Many *env* sites change amino acid preferences in the course of evolution

As a test case, we applied the d-test to the evolution of the gp160 protein encoded by the *env* gene in HIV-1. This protein evolves under recurrent positive selection [28,29], suggesting that site-specific amino acid propensities change between strains. We built a phylogenetic tree of 3789 sequences belonging to nine viral subtypes (see Materials and methods). As focal nodes, we considered the two strains tested in DMS experiments (see below): BF520 strain of the subtype A and LAI strain of the subtype B.

Among the 3491 amino acids with sufficient variability for our analysis (testable amino acids, see Materials and methods), we detected 59 variable fitness amino acids under the $\alpha = 0.01$ significance cutoff, distributed over 46 of the 724 sites of gp160 (Fig 2).

As expected, substitutions giving rise to the variable fitness amino acid were positioned on the phylogeny closer to the species for which they were proximal: the phylogenetic distance to that species comprised, on average, 0.87 (range: 0.57 to 0.99, standard deviation = 0.097) of that expected randomly. They also were positioned more remotely from the species for which they were distal: the phylogenetic distance to that species comprised, on average, 1.19 (range: 1.01 to 1.91, standard deviation = 0.19) of that expected randomly.

Of these 59 amino acids, 36 amino acids at 34 sites were proximal to the BF520 strain and distal from the LAI strain; and 23 amino acids at 23 sites were proximal to the LAI strain and distal from the BF520 strain. At 11 sites, changes in fitness were reciprocal: some amino acids were proximal to the BF520 strain and distal from the LAI strain, while others were proximal to the LAI strain and distal from the BF520 strain.

The sites carrying variable fitness amino acids were found at all domains and structural regions of gp120 and gp41, and were not biased towards or away from any functional domain (Fisher's exact test, mean two-sided p-value for functional domains > 0.6).

## Method validation with simulations and data

To understand the power of the d-test to identify amino acids with variable fitness, we used SELVa [30] to simulate the evolution of individual amino acid sites under a variable fitness landscape. SELVa produces forward-in-time simulations of sequence evolution along the phylogenetic tree provided as input, under non-uniform probabilities of substitutions for different amino acid pairs which are determined by the corresponding selection coefficients. In our model, substitution rates between amino acids were determined by their relative fitness, which

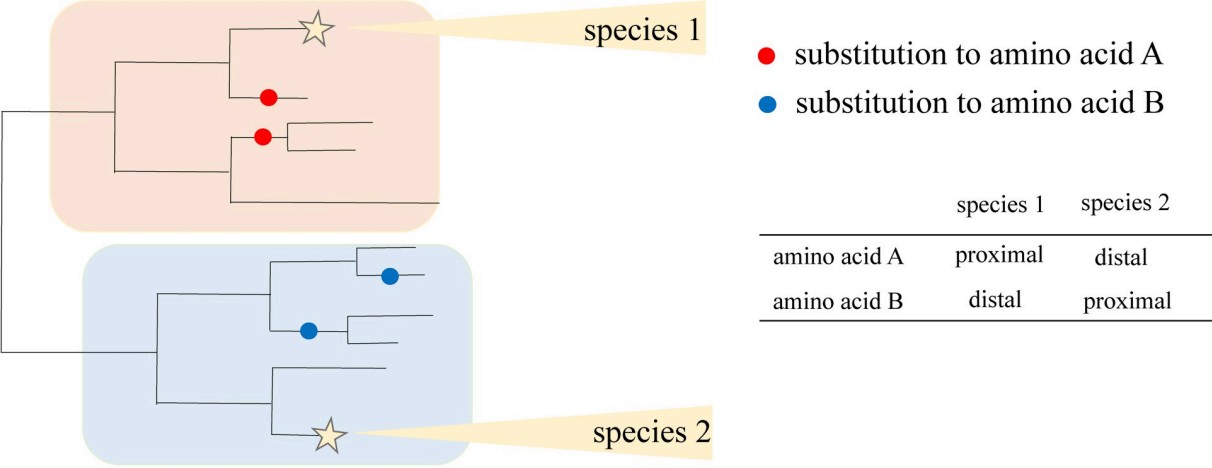

**Fig 1. Schematic representation of the approach for inference of changes in fitness of individual amino acids.** Both A and B are variable fitness amino acids when species 1 and 2 are chosen as focal nodes, because each of them is proximal to one species and distal from the other. By contrast, A is distal, and B is proximal, for both species in the comparison of species 2 and 3, so neither A nor B is identified as a variable fitness amino acid in this comparison.

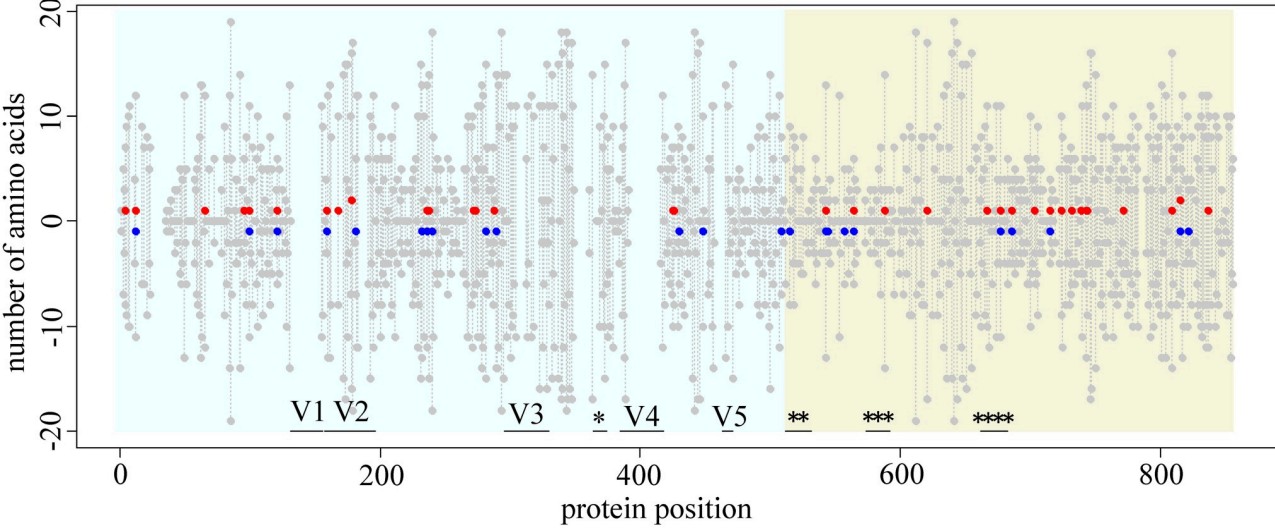

**Fig 2. Amino acid sites carrying variable fitness amino acids in gp160 protein of HIV-1.** For each position of the amino acid within the protein (horizontal axis), the red dot shows the number of amino acids that are simultaneously proximal to the BF520 strain and distal from the LAI strain; and the blue dot, the number of amino acids that are simultaneously proximal to the LAI strain and distal from the BF520 strain, multiplied by -1. Grey dots represent the number of testable amino acids (see Materials and methods). The positions without any dots are sites filtered out due to alignment ambiguity. Vertical grey lines connect dots for the same site. Blue background, gp120; green background, gp41. Variable loops V1-V5 and known functional domains are marked by horizontal black lines: *, CD4-binding loop; **, fusion peptide; ***, immunosuppressive region; ****, membrane proximal external region (MPER). Numbering of protein positions is hxb2-based.

could differ between tree branches. Specifically, we assumed that at a given site $a$ and at each position of the tree, exactly one of the 20 amino acids, $AA_k$, is favored by selection and has the scaled fitness $F_a^{AA_k} = 2N_e f_a^{AA_k} = X > 1$, where $N_e$ is the effective population size; while the remaining 19 amino acids have $F_a^{AA_{i \neq k}} = 1$. The $AA_k \longrightarrow AA_{i \neq k}$ substitutions then occur at rate $\frac{1-X}{1-e^{X-1}}$, while the $AA_{i \neq k} \longrightarrow AA_k$ substitutions occur at rate $\frac{X-1}{1-e^{1-X}}$ [31]. To model fitness variability, we assumed that selection changes exactly once at a predefined phylogenetic branch $b$; at this point, the identity of the newly favored amino acid with the fitness $X$, $AA_l$, is picked out of the remaining 19 amino acids randomly, and all remaining amino acids get fitness 1. In the process, the fitness of amino acids $AA_k$ and $AA_l$ has changed, while the fitness of the remaining 18 variants has remained constant.

We asked if the d-test could retrieve the identity of $AA_k$ and $AA_l$ from simulated protein sequences. For simulation, we used the reconstructed phylogenetic tree of the gp160 protein encoded by the *env* gene of HIV-1 (see Materials and methods). The tree contains 3300 tips and has two large clades corresponding to HIV-1 subtypes B and C (Fig A in S1 File). We assumed that selection has changed in the last common ancestor (LCA) of subtype C. To evaluate the dependency of the method performance on sampling density, we additionally ran the simulation for two artificial phylogenetic trees constructed on the basis of the *env* tree: "sparse tree" with roughly half of the tips and "dense tree" with twice as many tips as the original tree (see Materials and methods).

Overall, our simulations indicate that the d-test can detect the variable fitness amino acids with moderate sensitivity and very high specificity (Fig 3 (upper row) and Table A in S2 File). We find that under moderate selection ($X = 4$ to 9), we correctly identify on average 8%, 43% and 75% of amino acids for which fitness has changed in sparse, original and dense trees, respectively. Under stronger selection ($X = 10$), many sites are monomorphic, and the sensitivity of our method decreases, constituting 24% for the dense tree. Under weaker selection ($X = 3$), the power of the method also drops, probably because changes in selection of such a low magnitude are masked by the noise associated with drift and tree topology. The percent of amino acids with constant fitness that are identified as positives is low for all trees, never exceeding 0.6%.

As the d-test relies, for each amino acid, on comparing the number of substitutions giving rise to this amino acid and to other amino acids between segments of the phylogenetic tree, it is not applicable if either of these numbers is very low. The set of amino acids to which the d-test can be applied can be identified from the data (see Materials and methods); we refer to them as testable amino acids. When only testable amino acids are considered, the specificity of the test and its sensitivity for weak and moderate selection remain the same, while its sensitivity for strong selection increases substantially (48% for the dense tree; Fig 3 (bottom row)); this is because we no longer consider the sites where a variant is fixed in the entire clade where it is preferred.

To study how the performance of the d-test depends on the peculiarities of the dataset, we performed additional simulations. First, to study its dependence on the rate of evolution, we again simulated evolution along the *env* tree, but this time decreased or increased the tree height by multiplying all branch lengths by 0.25, 0.5, 2 or 4 (Fig C in S1 File). As was the case for the number of terminal branches, sensitivity of the test substantially increased in trees with longer branches, while specificity remained high for all trees (Table B in S2 File and Fig C in S1 File).

Second, we studied how the performance of the d-test depends on the position of the point where fitness shift occurs on the phylogenetic tree (Fig D (A) in S1 File). We found that this position is important (Table C in S2 File). This is because the d-test assumes that the part of

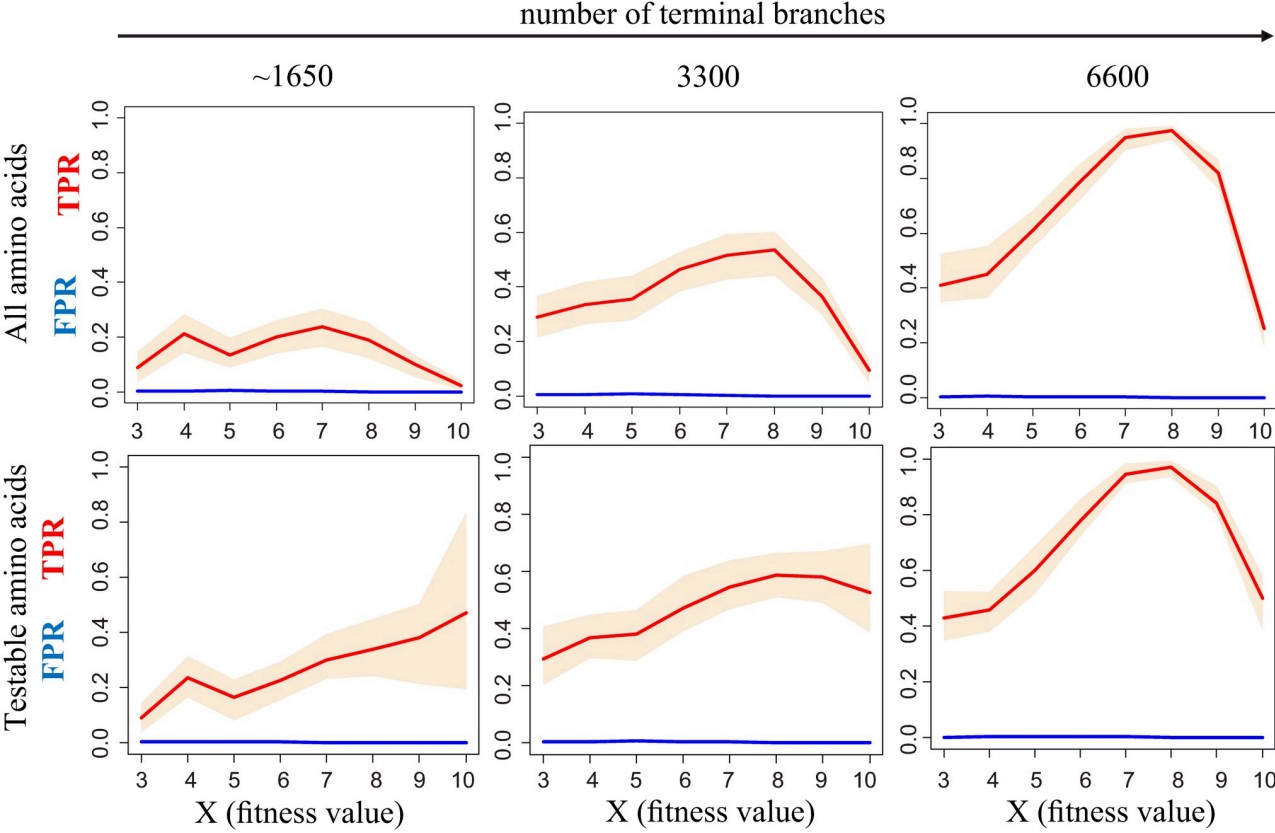

**Fig 3. Performance of the d-test for "sparse" (~1650 tips), original (3300 tips) and "dense" (6600 tips) phylogenetic trees in simulations for different scaled fitness values of the preferred variant.** Red line, median True Positive Rate (TPR); blue line, median False Positive Rate (FPR). 90% confidence bands were obtained by random subsampling of 1000 amino acids in 100 trials. Upper row: all amino acids are considered, bottom row– only testable amino acids are considered. A zoom-in for low FPR values is shown in Fig B in S1 File.

the phylogeny that is close to the focal node has the same fitness vector as the focal node. When this assumption is violated, the performance of the d-test decreases. This can happen, for example, if one or both of the focal species neighbors many species with a different fitness vector than itself (Fig E in S1 File).

Third, we asked how the performance of the d-test depends on the exact fitness spectrum of variants at the site. Overall, we found that it performs well for different fitness vectors. Specifically, the specificity remains high when several amino acids change their fitness, or when there are two preferred amino acids at a site and only one of them changes its identity in a shift point (Table C in S2 File).

Fourth, we asked how the performance of the d-test depends on the topology of the tree. For this, in addition to the *env* gene, we performed simulations over the phylogeny of mito-chondrial proteins of Metazoans and Fungi obtained from [20]. This tree includes several well-separated large clades, as opposed to just two such clades in the *env* tree (Fig D (B) in S1 File). We assumed that the shift has occurred in the beginning of the clade of fishes, and considered zebrafish and human as two focal species. In this analysis, the sensitivity of the method was very low (Fig F in S1 File and Table D in S2 File). However, when we repeated the analysis using only a subtree of vertebrates (Fig D (C) in S1 File), sensitivity substantially increased, reaching 20% for X = 7. The sensitivity further increased achieving 30% for X = 7 when the significance threshold was relaxed from 0.01 to 0.05, with almost no loss in specificity (99.998%

vs. 99.9887% for X = 7; Fig F in S1 File and Table D in S2 File). This analysis shows that the existence of clades that are distant from both focal points may swamp the phylogenetic signal (Fig E (D) in S1 File).

Overall, while the sensitivity of the d-test strongly depends on peculiarities of the phylogeny, it tends to be high under moderate selection strengths. The specificity of the test was high in all simulations.

Having established that the d-test performs well in simulations, we studied the robustness of the d-test when applied to real data. For gp160, the sets of variable fitness amino acids strongly intersected between ten randomly chosen pairs of focal strains from two viral subtypes as well as between ten trials with random subsets of 50% of strains (S3 File).

Next, we compared the results of the d-test with those of two recently developed methods that estimate the effect of amino acid mutations in a given sequence using multiple sequence alignment: GEMME [32] and De Mask [33]. For amino acids except the one which currently occupies the site, these methods compute a score which represents the predicted fitness effect of a substitution to this amino acid in the considered species. Therefore, the difference between the scores of an amino acid in two focal strains reflects the difference between these strains in its conferred fitness. Both these methods account for the identities of the amino acids, so estimating their performance in SELVa simulations is inappropriate for them (for GEMME, see Fig G in S1 File). Instead, we correlated the results of GEMME and De Mask with those of the d-test for the real-life data. To obtain a comparable measure for the d-test, for each amino acid, we calculated the differences between the standard scores of deviations of the d-statistic from the null expectation in the two focal strains. We obtained significant, albeit rather weak, correlations of fitness differences obtained by our method with those of GEMME (p < 2.2e-16, rho = 0.24) and De Mask (p = 7.0e-16, rho = 0.11) scores.

Finally, we checked the benefit of accounting for the phylogenetic tree by comparing our results with those of a simple baseline test based on differences in prevalence of an amino acid in the two subtypes (prevalence-based test, see Materials and methods). Although the sensitivity of this simple method was high for all X, its specificity was below 80% (Fig H in S1 File).

## Phylogenetically inferred changes in fitness between strains match those observed in DMS experiments

We compared the changes in phylogenetically inferred amino acid preferences between the two strains to those inferred experimentally by DMS [25,26]. To estimate the changes in DMS preferences, for each amino acid in each site, we measured the ratio $r = \pi_x/\pi_y$, where $\pi_x$ is the DMS preference for this amino acid in the strain for which it is phylogenetically more proximal, and $\pi_y$ is the preference for it in the strain for which it is more distal. As a control, we analyzed the distribution of $r$ between the two strains for invariable fitness amino acids.

The values of $r$ were significantly higher for the variable fitness amino acids than in the control (one-tailed Wilcoxon's test, p = 0.005; Fig 4), indicating that the phylogenetically and experimentally inferred variable fitness amino acids are correlated. This trend is pronounced both for amino acids that conferred higher fitness in BF520 and in LAI strains (Fig 4). It is mainly driven by amino acids with modest preferences ($\pi < 0.2$; Fig 4B), which are not the highest-ranking amino acids in DMS experiments in either of the strains (Fig 4C).

As the gp160 evolves fast, and its alignment contains many gaps, we checked whether the agreement with the experiment would remain after more stringent filtering of sites. When we removed from consideration all amino acid sites having at least one gap within five positions more than in 5% of the dataset, the values of $r$ were again higher for the variable fitness amino

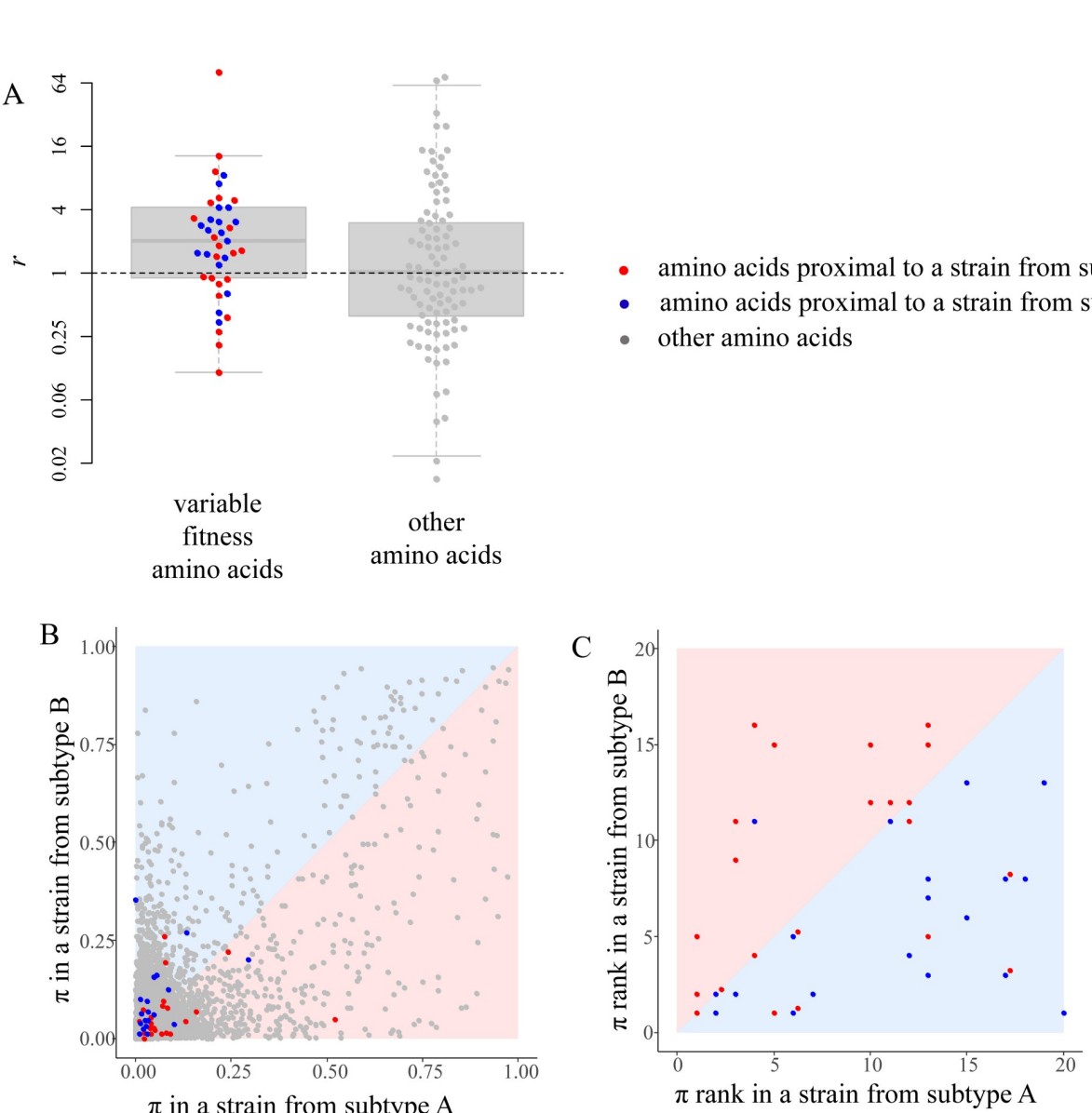

**Fig 4. A**, The ratio of DMS preferences π for an amino acid between the strain for which this amino acid is phylogenetically more proximal and the strain for which it is more distal. Amino acids with significant (left, p<0.01 for both strains) or insignificant (right, p>0.1 for both strains) changes in conferred fitness between strains are shown; bold line is the median, box borders represent the inter quartile range (IQR) between the first (Q1) and the third (Q3) quartiles, whiskers are Q1–1.5*IQR and Q3 + 1.5*IQR intervals, outliers are not shown; boxplots are made using all data; each variable amino acid significantly proximal to a strain from subtype A (BF520) or a strain from subtype B (LAI) and a subsample of 100 other amino acids are shown as red, blue and grey dots respectively. The dashed line indicates no change in preferences (*r* = 1). **B, C**, Experimentally measured fitness π (**B**) or fitness rank (**C**, with rank 1 corresponding to the fittest amino acid) of each amino acid in the two strains of HIV-1. Grey dots, amino acids that have been observed at least in one strain in our phylogenetic tree; red, variable fitness amino acids that are proximal to a strain from subtype A (BF520); blue, variable fitness amino acids that are proximal to a strain from subtype B (LAI); colored areas indicate the expected positions for dots of corresponding color.

acids than in the control, although the significance was moderate as only 12 variable fitness amino acids were left after this filtering (one-tailed Wilcoxon's test, p = 0.04).

For variable fitness amino acids, there is a significant correlation between *r* and the extent to which substitutions into one of the amino acids are biased towards a certain part of the phylogeny, measured as the difference between the standard scores of d-statistic between the subtypes for which this amino acid was proximal and distal (Spearman correlation test, p = 0.04, rho = 0.24). This suggests that variable fitness amino acids with the highest effect size are the most reliable.

## Most sites with variable fitness amino acids show no evidence of recurrent positive selection

Changes in fitness conferred by an allele can trigger episodes of positive selection. To study the association between these processes, we identified sites under recurrent positive selection, as evidenced by an above-neutral ratio of nonsynonymous to synonymous substitutions (dN/dS) estimated by codeml ($\omega > 1$) in subtype A, subtype B or both. Among the 46 sites carrying variable fitness amino acids, only 20 (43%) have experienced positive selection in subtypes A or B, despite the fact that the power of both the site-specific dN/dS test for positive selection and our test increases with the number of substitutions. An even lower overlap was observed with the sets of positively selected sites described in previous studies ([29]:17%, [34]:9%). Although there was an excess of positive selection among sites with variable fitness amino acids (Materials and methods; Fisher's exact test, two-sided p < 0.001, Table 1), the fact that no such selection was detected in over a half of variable sites suggests that these two phenomena are distinct. Furthermore, *r* was not higher for variable fitness amino acids at positively selected sites than at sites without any signal of positive selection (one-tailed Wilcoxon's test, p = 0.09).

A closer examination of the data helps understand the interplay between the dN/dS ratio and fitness variability. For example, according to the DMS data, Y159 is strongly preferred in a strain from subtype A, while F159 is strongly preferred in a strain from subtype B. In our phylogenetic tree-based analysis, Y159 is a variable fitness amino acid proximal to the strain from subtype A. At site 159, codeml has detected positive selection in subtype A ($\omega = 3.47$), but not in subtype B ($\omega = 0.10$). All substitutions to Y are from F, and within subtype A, F159Y

**Table 1. Sites with variable fitness amino acids and recurrently positively selected sites.**

| | Sites with variable fitness amino acids | Sites without variable fitness amino acids | p-value* |
|---|---|---|---|
| **Our analysis** | | | |
| Positively selected sites | 20 | 71 | <0.001 |
| Other sites | 26 | 422 | |
| [34] | | | |
| Positively selected sites | 4 | 21 | 0.26 |
| Other sites | 42 | 472 | |
| [29] | | | |
| Positively selected sites | 8 | 37 | 0.04 |
| Other sites | 38 | 456 | |

Only sites with testable amino acids are included.

*2-tailed Fisher's exact test

substitutions constitute nearly all (81%) observed substitutions at site 159, so the signal of positive selection picked up by codeml has been created by these substitutions, implying recurrent adaptation (Fig 5A).

By contrast, while S273 was designated a variable fitness amino acid in our analysis, no positive selection has been detected at site 273 by codeml. All substitutions to S are from R, and R273S substitutions constitute only 5.7% of all substitutions at this site in subtype B, while they account for 46.5% of substitutions in the remaining subtypes (including subtype A). Consistently, S273 confers higher fitness in DMS experiments in subtype A strain than in subtype B strain. Still, even within subtype A, S273 is just one of the several variants with near-equal fitness (Fig 5B). Therefore, the increased frequency of R273S substitutions in subtype A is likely due to relaxation of negative selection against 273S, rather than positive selection in its favor.

## Differences in fitness constraints between 3 subtypes

Our approach can be extended to detect amino acids with fitness differing in a predetermined pattern between multiple regions of the phylogenetic tree. To illustrate this, we study the amino acids that change the conferred fitness between three strains corresponding to the three major subtypes of HIV-1, A, B and C, which are the most prevalent subtypes in Asia, Northern America/Europe and Africa, respectively. Besides the 59 variable fitness amino acids detected at 46 different sites in the A-B pairwise analysis (see above), we also found 268 additional amino acids that had changed fitness between C and at least one of the subtypes A and B. Overall, we detected 7 amino acids at 5 sites in the A-C comparison, and 285 amino acids at 152 sites in the B-C comparison. These amino acids are listed in S1 Table; the web script that lists the variable fitness amino acids between subtypes is available at http://makarich.fbb.msu.ru/galkaklink/hiv_landscape/.

## Changes in amino acid propensies may be associated with changes in immunogenicity

To illustrate the potential biological basis of the observed changes in amino acid propensies, we studied how broadly neutralizing antibodies (bnAbs) bind the epitopic sites of gp160. We hypothesized that changes in binding between the major HIV-1 subtypes may be associated with changes in propensities at corresponding sites.

To test this, we considered 9 bnAbs; among those, 6 target the amino acids of gp120 (including VRC01, b12, NIH45-46 which target the CD4-binding site, PG9 which has an epitope in the V1/V2 loop region, and PGT121 and PGT128 that contact the outer domain of gp120 and the V3 loop base); 1 (2G12) targets glycans in the outer domain of gp120; and 2 (2F5 and 4E10) contact with gp41 (Fig 6A). For each bnAb, we asked if it differed in inhibition capacity between subtypes. For this, we compared its experimentally measured half maximal inhibitory concentration (IC50) values between subtypes [35,36]. For 5 of the 9 bnAbs, the differences were significant in at least one of the comparisons (Table E in S2 File).

We then mapped the sites carrying variable fitness amino acids in the A-B (46 sites) or the B-C (152 sites) comparison onto the three-dimensional structure of the gp120/gp41 trimer and its complexes with bnAbs. Among these sites, 7 and 28 respectively were epitopic, i.e., were involved in binding with bnAbs according to [37]. (In the A-C comparison, only 5 sites carried variable fitness amino acids, and only one of them was epitopic; these were not analyzed further).

For many of the bnAbs that substantially differed in binding between two subtypes, we found that the associated epitope carried some sites with variable fitness amino acids between these subtypes (Table E in S2 File). For example, the two epitopes that are responsible for

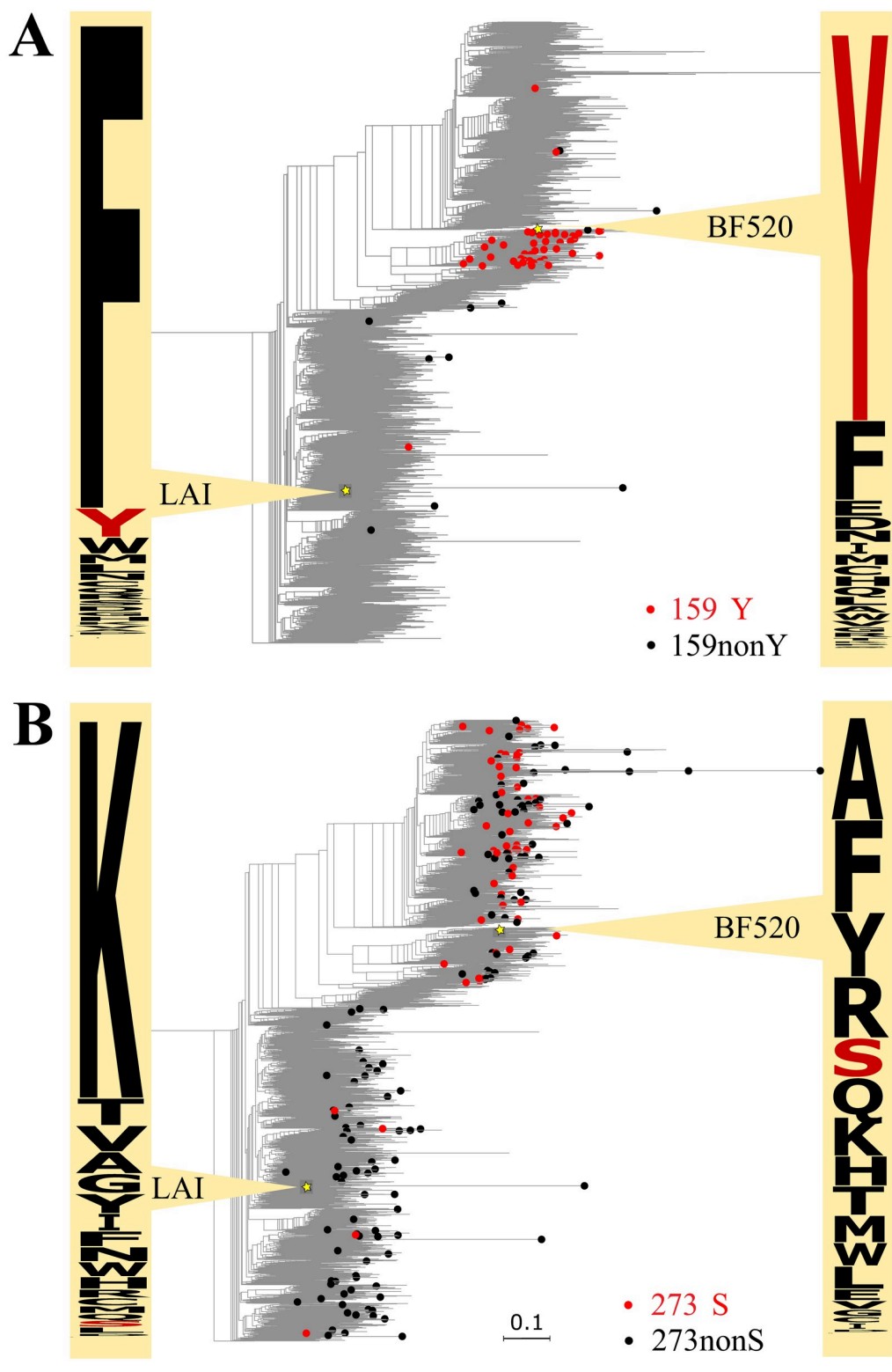

**Fig 5. Substitutions to variable fitness amino acids (red) and to other amino acids (black) at sites 159 (A) and 273 (B) of gp-160, and amino acid propensities in DMS experiments in strains BF520 and LAI.** Phylogenetic distances are in numbers of amino acid substitutions per site. Numbering of protein positions is hxb2-based. Logos of experimentally derived amino acid propensities were built with Weblogo (https://weblogo.berkeley.edu/logo.cgi).

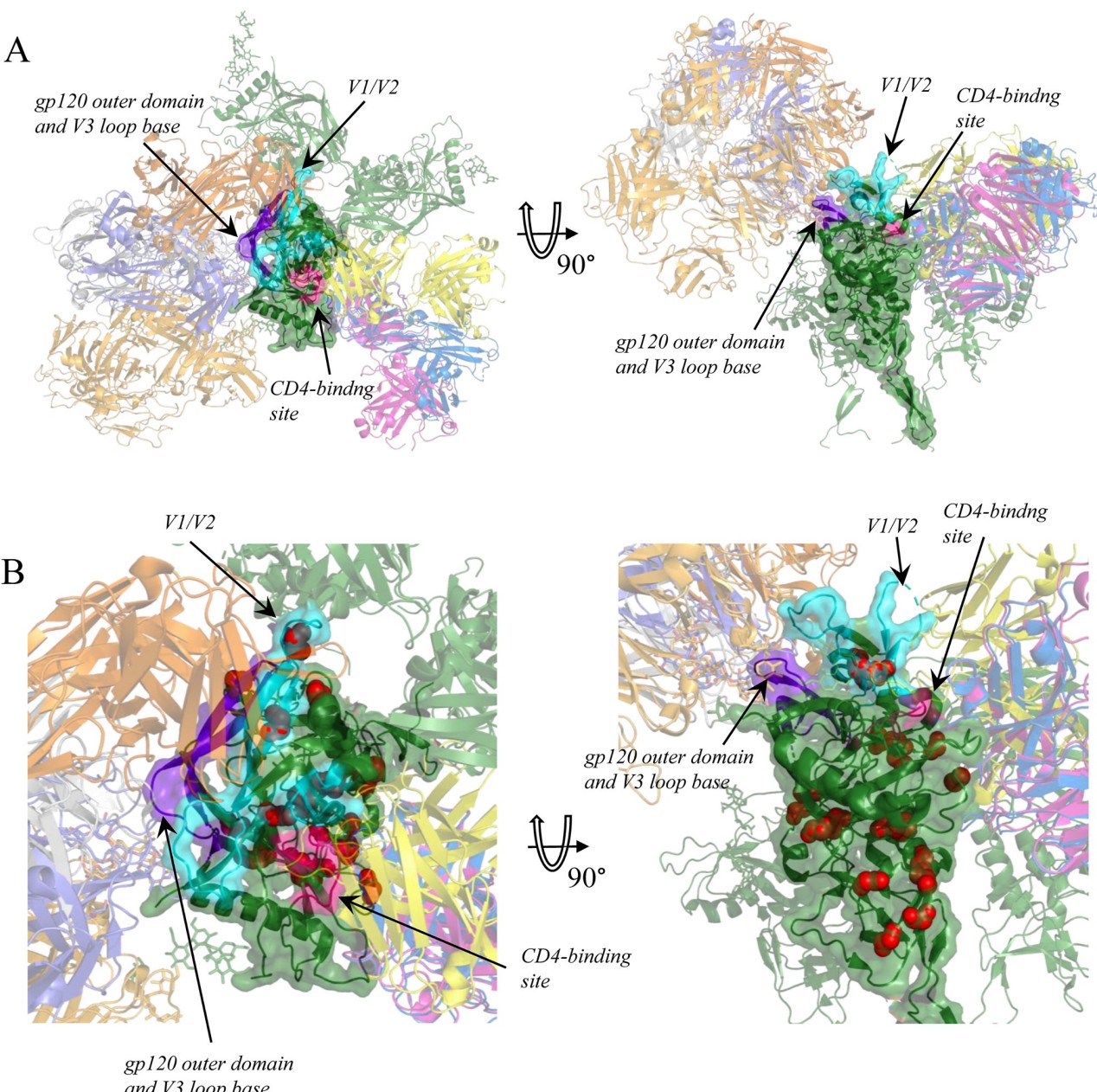

**Fig 6. A**, Overview of the structure of the gp120 trimer and its complexes with bnAbs (see Materials and methods). gp120 is colored green and shown as cartoons, one subunit is also shown as a semi-translucent surface. bnAbs are in various color and shown as cartoons. Glycan molecules are shown as sticks. bnAbs binding only one gp120 subunit are shown for clarity. **B**, Close-up of one gp120 subunit, with epitopes signed; sites with variable fitness amino acids between subtypes A and B shown as red spheres. Epitopic sites are defined according to [37]; CD4-binding site is colored pink, V1/V2 loop region cyan, and V3 loop region purple.

differential binding between subtypes A and B (those in the V1/V2 region and the CD4-binding site) both carry amino acids with variable fitness between these two subtypes, while the remaining of the gp120 epitopes (the one in the V3 loop) carries no such amino acids and experiences the same level of binding between the two subtypes (Fig 6B).

This suggests that shifts in fitness landscapes of epitopic sites may be associated with changes of antigenic properties of these epitopes. Remarkably, in 16 out of 18 sites that carry

variable fitness amino acids and reside in epitopes differentially bound by antibodies, the amino acid that actually occupies the subtype consensus remains the same between the two subtypes, despite changes in propensities (Table F in S2 File). This suggests that the differences in binding may be associated with changes in relative fitness of amino acids in epitopic sites that did not lead to change in identity of the most prevalent amino acid. For example, an increase of affinity of an epitope due to a substitution in one of its sites may change the relative fitness of amino acids at other sites of this epitope, but if selection against increased binding is not strong enough, this may not lead to a change of the consensus amino acid.

## Variable fitness amino acids in HA of Influenza A

We applied the d-test to the HA gene of H1 and H3 subtypes of human influenza A and detected 3 variable fitness amino acids under the $\alpha$ = 0.01 significance cutoff. Although statistically our results did not significantly correlate with the experimental fitness measures from [27] (one-tailed Wilcoxon test, p = 0.08), all three amino acids had higher experimental fitness in strains for which they were proximal than in those for which they were distal. Under a higher significance cutoff ($\alpha$ = 0.05), we detected 9 variable fitness amino acids at 7 sites (Fig 7).

The identities of these 9 amino acids were again consistent with the DMS experiment: amino acids with an excess of substitutions into them in the phylogenetic neighborhood of one of the strains conferred higher fitness at this strain than in the other strain (one-tailed Wilcoxon's test, p = 0.02; Fig 8).

For some of the sites, changes in fitness could be interpreted from the functional perspective. For example, site 88 (A/Puerto Rico/8/1934 numbering) is antigenic in H1, but not in H3 [38]. We suggest that variability in fitness of amino acid L that we found reflects differences in the amino acid propensities at this site between the subtype where it is epitopic (H1; Fig 9A) and subtype where it is not epitopic (H3; Fig 9B). Our suggestion is supported by the fact that 7 amino acids can be observed at this site in H1, but only 2 in H3.

Another example is site 202 (in A/Perth/16/2009 numbering) that was previously shown to be antigenic in H3, but not in the H1 subtype [39]. At this site, S is the ancestral amino acid for both H1 and H3 clades. Both our analysis and the DMS experiment suggest that P202 confers higher relative fitness in H1 than in H3. Four S202P substitutions have occurred in the H1 clade, and 90% of the H1 isolates contain P202. Conversely, no substitutions into P have occurred in the H3 clade, and 95% of H3 isolates carry G202, which has originated from S in a total of 3 substitutions. The observed differences in substitution patterns might reflect the differences in the function of site 202 between the two viral subtypes (Fig 9).

## Discussion

Patterns of substitutions in the course of evolution of a protein are shaped by the selection pressure experienced by it, and changes in this pressure may lead to non-uniformity of substitution rates between evolving lineages. While the overall non-stationarity of evolutionary rate [heterotachy; 40–42] as well as of individual substitution types [heteropecilly; 19,20,43] have been used to infer changes in underlying selection, these approaches require pooling multiple sites to obtain a statistical signal. Therefore, they are unable to pick up changes in amino acid propensities within individual sites, especially if the directions of these changes were discordant between sites.

Large sequence datasets with inferable phylogenetic relationships can be harvested to increase the resolution of these methods. Here, we develop an approach to infer, for individual

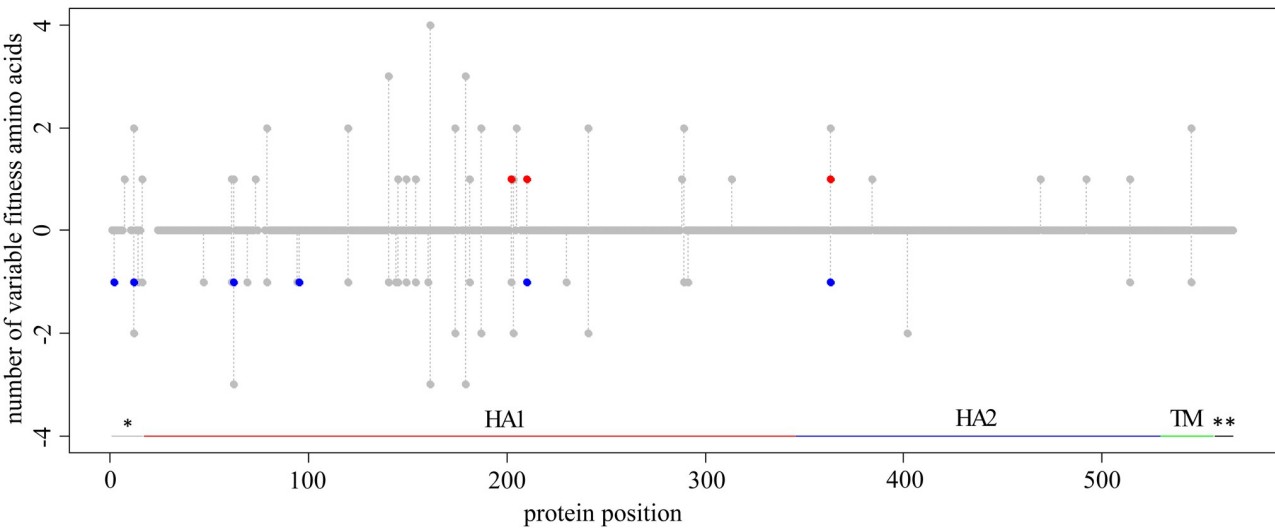

**Fig 7. Amino acid sites carrying variable fitness amino acids in HA of influenza A virus.** For each protein position (horizontal axis), the red dot shows the number of amino acids that are simultaneously proximal to the H1 strain and distal from the H3 strain, and the blue dot, the number of amino acids that are simultaneously proximal to the H3 strain and distal from the H1 strain, multiplied by -1. Grey dots represent the number of testable amino acids (see Materials and methods). The positions without any dots are sites filtered out due to alignment ambiguity. Vertical grey lines connect dots for the same site. Protein domains are marked by horizontal lines: *, signal peptide; HA1, HA1 ectodomain; HA2, HA2 ectodomain; TM, transmembrane domain; **, cytoplasmic tail; numeration of protein positions is A/Perth/16/2009 based.

amino acids at individual sites, the changes in relative fitness that they confer between two related lineages.

Our method (the "d-test") makes use of changes in relative frequencies of convergent (or parallel) and divergent substitutions between evolving lineages. While the high frequency of substitutions giving rise to the same amino acid suggests a high fitness of that amino acid, usage of divergent substitutions provides an internal control, making the d-test robust and insensitive to other types of non-stationarity. In particular, first, it is not sensitive to differences in the overall substitution rate (across sites and mutation types) between lineages. Second, as it considers each site individually, it is not sensitive to differences in substitution rates or patterns between sites. Third, as it controls for the identity of the ancestral amino acids, it is not sensitive to differences in rates of different types of substitutions (e.g., transitions vs. transversions) within a site that are conserved throughout the considered phylogeny. Fourth, for the same reason, it is not sensitive to changes in the rate of the substitutions of the same type that are caused by the changes in the prevalence of a particular ancestral amino acid [19].

Still, our method has some limitations, most of which have been discussed previously [19]. First, as it is based on the analysis of phylogenetic distribution of amino acid substitutions, its power is limited by the number of substitutions observed, which in turn depends on mutation rates, the tree shape and the selection regime. In particular, it is inadequate for detection of changes in amino acid fitness between closely related species or at slowly evolving sites. Here, we explicitly study the power of the d-test for different selection strengths. Simulations show that our method works best within a window of selection coefficients corresponding to moderate selection. This is to be expected. If selection is too weak, both weakly advantageous and weakly deleterious substitutions occur throughout the phylogeny, making our approach impotent. Conversely, if selection is very strong, the number of variable sites becomes low: the favored amino acid is usually nearly fixed in one of the clades, so that there are no substitutions giving rise to it to analyze. In particular, the d-test is inadequate for detection of selection in

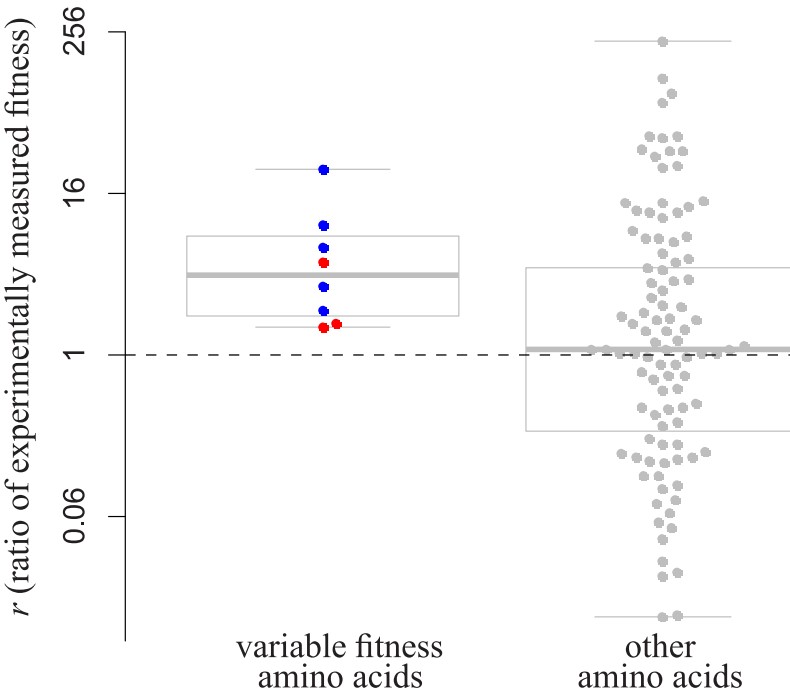

**Fig 8. The ratio between experimentally measured amino acid preferences of the strain for which an amino acid is more proximal and the strain for which it is more distal in HA of influenza A.** Amino acids with significant (left, $p<0.05$ for both strains) or insignificant (right, $p>0.1$ for both strains) changes in the phylogenetic analysis are shown; notations are the same as in Fig 4.

favor of a highly conserved ancestral variant, since multiple substitutions into this variant are unlikely. While the prevalence of moderate selection coefficients in proteins varies substantially between studies [44,45], it is considerable in some datasets, including viral proteins [46].

Furthermore, the power of the d-test increases when sequences of more species are available (Fig 2). This happens for two reasons: first, an increase in the number of substitutions gives more power; second, even with the same number of substitutions at a site, the power is higher for sites with more distinct ancestral amino acids, as at such sites, there are more combinations of substitutions in the null distribution, and thus a lower minimal p-value can be obtained.

Second, the d-test is based on the assumption that the similarity between the substitution patterns (which are determined by amino acid propensities) is associated with the distance between the lineages in the phylogenetic tree; in other words, more closely related lineages are more likely to share the amino acid propensities. While this assumption probably holds on average, it can be easily violated in specific cases: for example, propensities can be shared with more remote lineages while they can be different from those in closer ones, depending on the phylogenetic position of the considered species and details of the tree shape (Fig D in S1 File). When this assumption is violated, the performance of the test will drop (Table D in S2 File). This problem can be alleviated by only including in the analysis the minimal clade containing the two focal species, and excluding more remote species that can confound the signal (Table D in S2 File).

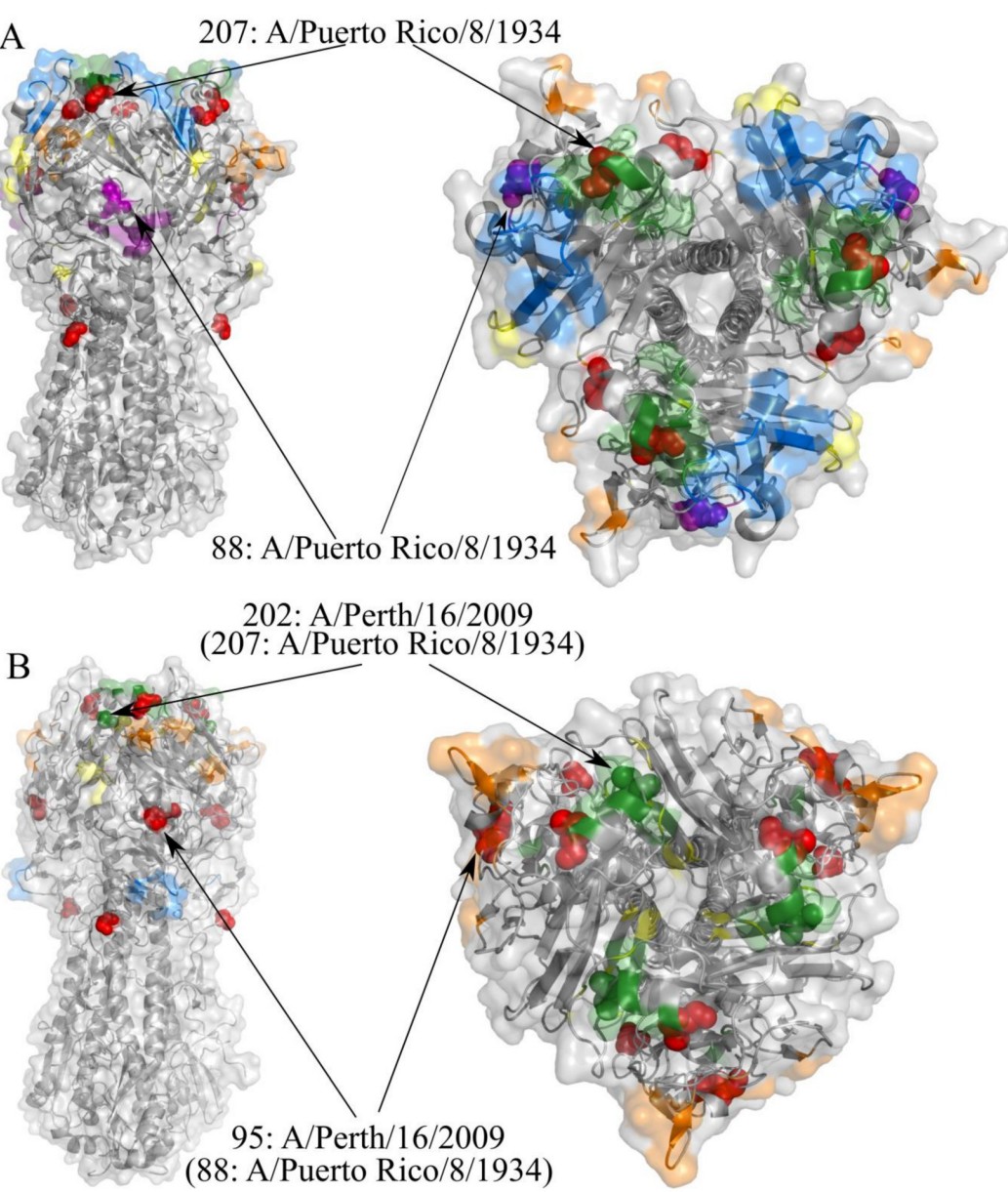

**Fig 9. Variable fitness sites in the structure of the Influenza hemagglutinin.** Sites with variable fitness amino acids are shown as red spheres except when they are located in one of the antigenic sites. **A**, Subtype H1 trimer (PDB id 1ru7). Antigenic sites were mapped based on [38]: Ca1 is shown in yellow, Ca2 in orange, Cb in magenta, Sa in blue, Sb in green. **B**, Subtype H3 trimer (PDB id 2viu). Antigenic sites were mapped based on [39]: A is shown in orange, B in green, C in blue, D in yellow.

Third, our test detects any change in the frequency of an individual substitution at a site independent of its cause. While at the amino acid level, such changes are probably always caused by changes in amino acid propensities, if a similar approach is applied to nucleotide substitutions, rate changes can also occur due to differences in mutation biases or in non-amino-acid level selection, e.g. selection for nucleotide composition. In theory, synonymous sites can be used to control for non-selective changes in substitution biases, although this is not straightforward for viruses where many of such sites can be constrained by selection on

structure or overlapping reading frames. However, mutation spectra are similar between HIV-1 subtypes [47], strongly suggesting that the changes we detect are caused by differences in selection rather than mutation.

As the probability of a substitution to an amino acid depends on its $N_e$-scaled relative fitness, theoretically, the d-test could be sensitive to differences in $N_e$ between subtypes. We are unaware of any data showing systematic differences in $N_e$ between HIV or influenza A subtypes. More generally, changes in $N_e$ should affect all sites concordantly, and it is difficult to imagine a scenario whereby they will lead to changes in density of homoplasies of opposite directions between sites, as observed in our analysis.

Fourth, our method relies on the correctness of the inferred phylogeny. As significant, we only considered those sites that are reproduced in 10 out of 10 independently reconstructed phylogenetic trees, and therefore believe that our findings are robust to errors in overall shape of the gene tree. The importance of considering phylogenetic relationships between sequences in the alignment is confirmed by the higher performance of the d-test compared to a test based solely on differences in amino acid prevalence between subtypes (Fig H in S1 File). Also, for real-life gp-160 data, the values of $r$ for variable fitness amino acids inferred by the d-test were significantly higher than those for amino acids inferred by prevalence test (Wilcoxon rank sum test, p = 0.028; Fig I in S1 File), indicating that the phylogeny-based test agrees with the experimental data better than the prevalence-based test.

Incongruence between phylogenies of the gene and the considered site can also result from within-gene recombination. While we exclude known recombinants from analysis, some recombination events, particularly within-subtype ones, could have been undetected. To result in spurious clustering of homoplasies, such recombination would have to be recurrent (as each recombination event can lead to at most one spurious homoplasy), and to preferentially affect closely related lineages (as no clustering would be observed otherwise). However, it is unclear why random recombination would specifically favor amino acid variants that have high experimentally measured fitness in the corresponding clade.

To illustrate and experimentally validate our approach, we make use of two DMS datasets for viral surface proteins, in each of which the amino acid preferences have been measured experimentally for two different strains: the gp160 protein in two HIV-1 strains, and the HA protein in two influenza A strains. Overall, we find that the results of the phylogenetic test are consistent with experimental results. If a particular amino acid is gained more frequently by a species closely related to a particular strain, it also tends to confer higher fitness in experimental assays in that same strain.

We find that the inferred variable fitness amino acids mostly have modest experimentally measured preference π even in the strain for which they are proximal. There can be multiple reasons for this. First, our phylogenetic test is not guaranteed to assign high preference to the fittest amino acid at a site. In particular, it is only able to assign differential fitness to derived, but not ancestral, variants. For example, 159F confers high experimentally measured fitness in the gp160 of the LAI strain, but its fitness is lower in the BF520 strain (Fig 5A); still, it cannot be picked up by the phylogenetic test, as it is the ancestral variant, and the substitutions giving rise to it are rare in both strains (Fig 5A). As the variant that is much more fit than all other variants is likely to be conserved throughout the clade, the phylogenetic method is less likely to pick it up, compared to a variant with fitness comparable to that of at least one other variant.

Second, many of the fitness shifts between A and B subtypes of HIV-1 did not affect the fittest amino acid. Indeed, among the 42 amino acids with significant phylogenetically inferred fitness shifts between these subtypes, only 8 (19%) had the highest DMS-inferred fitness rank in either strain (Fig 4C). The remaining 34 amino acids were suboptimal in both strains. Therefore, the d-test allows to infer the changes in fitness between suboptimal amino acids,

which is relevant for understanding the overall fitness landscape governing the evolution of the protein.

An evolutionary model that incorporates the results of a DMS scan [ExpCM; 48] has been shown to outperform other substitution models, and, in particular, detect positive selection more robustly. This is achieved by obtaining the strength of functional constraint, indicative of negative selection, for individual sites from the DMS data [26,27]. In line with that model, the changes in substitution rates observed here can be caused by changes in the strength of negative selection between strains. Additionally, changes in amino acid preferences can trigger episodes of positive selection if the newly preferred amino acid is rare in the ancestral population. We find that less than a half of sites with variable fitness amino acids also had signatures of positive selection that could be detected by a standard dN/dS approach. Therefore, changes in site-specific fitness landscape are not necessarily coupled with strong positive selection favoring new optimal variants.

Our method can be applied to more than two focal nodes to find amino acids that have the same relative fitness in some lineages but change it in other lineages. We illustrate this using the three subtypes of HIV-1. As the number of strains substantially differs between these three clades (223, 2035 and 1265 for A, B and C correspondingly), the power of the d-test differs between them. Still, our results show that with enough data we are able to compare fitness of an amino acid between more than two clades. Importantly, unlike other methods for detection of changes in selection preferences, our test can be applied without prior identification of candidate clades differing in selection pressure [49–51]. Also, it makes no assumption regarding the model of epistatic interactions or fitness shift [49–52] which makes it resistant to overparameterization and reduces computations, allowing to use larger datasets and potentially capture more subtle changes in amino acid preferences. The d-test does not explicitly model epistasis between sites, and does not consider co-occurrences of amino acids in different positions of the protein. Such epistasis however would contribute to the detected signal, as more phylogenetically related samples are also more similar in sequence.

Our results were significantly correlated with those obtained by GEMME [32] and De Mask [33], two methods that estimate the effect of amino acid mutations on fitness, but this correlation was weak. These methods seem to have the highest power for different situations. GEMME and De Mask lean partially on physico-chemical similarity of amino acids and on site conservation. By contrast, the d-test does not depend on these properties, and may find differences in fitness effects of amino acids at sites that are not conservative in terms of GEMME or De Mask but that still may have important functions. Moreover, both GEMME and De Mask are unable to find fitness shifts for the amino acids that occupy the site in either of the focal species, whereas d-test can do that. This emphasizes the importance of using several different methodologies that can complement each other when looking for sites with variable fitness landscapes.

Predicting the effects of genetic variation on fitness is often based on the extent of conservation of a position [53–55], but functionally important residues are not necessarily conserved [56]. Our method can be used to distinguish between rapidly evolving sites with relaxed and variable constraints. For example, in gp160, there are 10 sites each with 18 different amino acids that originated at some branch (es) of the phylogenetic tree. While this is suggestive of complete neutrality of these sites, we find that, among these 10 sites, 5 carry variable fitness amino acids.

Despite the recent surge in DMS experiments that allow high-throughput fitness measurements, such experiments can still only be performed for a relatively few model systems, and are still impractical for large numbers of strains. Moreover, the fitness measurements in experimental conditions are not necessarily reflective of the selective pressure experienced by a

lineage over the course of its evolution [25]. By contrast, comparative genomics allows to assess the fitness differences that have shaped evolution for a large number of non-model species simultaneously.

## Materials and methods

### Inference of phylogenetic biases (principle of the d-test)

The d-test requires an amino acid alignment and a phylogenetic tree with reconstructed ancestral amino acid states. We define the phylogenetic distance $d$ between the focal nodes and each substitution into amino acid $A$ as the sum of branch lengths along the path between them on the phylogeny, assuming that a substitution occurs at the middle of the corresponding branch, and $\bar{d}$ as the mean of those distances. If the substitutions preferentially occur in the phylogenetic neighborhood of the focal nodes, $\bar{d}$ will be lower than expected. By contrast, if substitutions are underrepresented in the phylogenetic neighborhood of the focal nodes, $\bar{d}$ will be higher than expected. To obtain the null distribution of $\bar{d}$, we make use of the substitutions into non-$A$ amino acids. $\bar{d}$ depends both on the phylogenetic distribution of the amino acids ancestral to $A$ and on the number of substitutions into $A$ [19]. To control for this, we subsampled the substitutions from the same ancestral amino acids as follows. We listed all "ancestral" amino acids that were substituted by $A$ at least once. For each such ancestral amino acid, we then considered all substitutions into any ($A$ or non-$A$) amino acid, and subsampled, out of these substitutions, the number of substitutions equal to that of substitutions into $A$. We then pooled these substitutions across all ancestral amino acids together, and measured the mean phylogenetic distance $\tilde{d}$ for them. This subsampling was repeated 10,000 times to obtain the null distribution of $\bar{d}$.

The p-value for the bias of substitutions towards the focal nodes was defined as the percentile of $\bar{d}$ in this distribution, and for their bias away from the focal nodes, as 1 minus this value. We required p < α for proximal amino acids, and p > (1-α) for distal amino acids, where α is the significance threshold. An amino acid was considered "testable" (proximal or distal) if the minimal attainable p-value (i.e., the fraction of the values of $\tilde{d}$ in the most extreme bin) was lower than α. For each significance threshold α, we calculated the expected fraction of false positives (Q-value) as min (α*P)/V, where P is the number of testable amino acids at this α, and V is the number of significant amino acids at this α. We used α = 0.01 for gp160 throughout; the percent of expected false positives at this p-value is less than 12% (Fig J in S1 File). For HA, where the substitution rate is lower, which led to more than 50 times lower P at α = 0.01, we used α = 0.05, yielding the Q-value<62% (Fig J in S1 File).

Finally, for a pair of species 1 and 2, we defined an amino acid as "variable fitness amino acid" if it was proximal for one species and distal for the other one. All amino acids that were potentially significantly proximal for one species and potentially significantly distal for the other were considered as "testable amino acids".

To measure the magnitude of the effect, we calculated the standard score for $\bar{d}$, i.e., the number of standard deviations between the mean of the null distribution of $\bar{d}$ and the observed $\bar{d}$. For each amino acid, we define two scores: one for the first focal species and one for the second focal species. For variable fitness amino acids, the score is negative for a focal species for which the amino acid is proximal, and positive for the focal species for which it is distal. The magnitude of the effect is then defined as the difference between the distal and proximal scores.

All scripts needed to run the d-test along with the Manual can be found on GitHub: https://github.com/GalkaKlink/d-test.

## Prevalence-based test

This test is based solely on the alignment, and does not account for the phylogeny. For each amino acid at each site of the protein, the test statistic is calculated as the difference between the number of sequences that carry this amino acid in subtypes A and B normalized by its prevalence in both subtypes ((numberA-numberB)/(numberA+numberB)). The null distribution is obtained by reshuffling subtype labels between sequences in 1000 trials. If the value of the statistic is unexpectedly high given the significance threshold, the amino acid is considered to be preferable in subtype A, and if it is unexpectedly small, it is considered to be preferable in subtype B.

## Data

**gp-160.**    We downloaded 4241 gp160 DNA and protein sequences belonging to nine subtypes of HIV-1 from the Los Alamos HIV sequence database (http://www.hiv.lanl.gov/) using the following settings: alignment type = filtered web, organism = HIV-1/SIVcpz, region = Env, subtype = M group without recombinants, year: 2016. The subtypes were represented unequally, with the majority of sequences from subtypes B (>2000 isolates) and C (>1000 isolates). We discarded sequences with lengths not divisible by 3 and internal stop codons, leaving 3789 sequences. We then added two experimentally studied sequences LAI and BF520, and an outgroup type O (isolate_ANT70), for a total of 3792 sequences. Using Pal2Nal [57] and Mafft [58], we obtained amino acid alignments and codon-informed nucleotide sequence alignments and filtered out positions with more than 1% of gaps and ambiguous characters. Using nucleotide alignments, we built a maximum likelihood phylogenetic tree using RAxML under the GTRGAMMA model. We then optimized branch lengths on the basis of the amino acid alignment under the PROTGAMMAGTR model [59] and reconstructed ancestral states with codeml program of the PAML package [60].

The analysis was performed on amino acid alignments, using LAI or BF520 as focal nodes. The initial length of these samples was 852 and 861 amino acids, respectively. The initial protein alignment included 2488 positions, and 724 positions containing on average 8.6 different amino acids per site remained after filtering (with 3.5, 6.6 and 6.3 amino acids per site in subtypes A, B and C, respectively). Among those, the DMS results for both focal strains were available for 562 sites.

To ensure that our results are robust to uncertainty of phylogenetic reconstruction, for all analyses involving one "proximal-distal" comparison, we constructed maximum likelihood phylogenetic trees on the basis of the same alignment in 10 independent runs, identified the variable fitness amino acids for each tree, and performed downstream analyses with just those amino acids that had significant shifts in fitness in all 10 replicates.

Functional regions were defined as annotated in UniProt [61; entry P04578]. We obtained solvent accessibility data for the LAI strain from [25]. To define buried and exposed residues, we used the threshold relative solvent accessibility = 0.25.

**Hemagglutinin.**    We obtained 2238 and 50,109 DNA sequences of HA from seasonal H1N1 and H3N2 influenza A subtypes correspondingly from the GISAID database [62]. We discarded sequences shorter than 1680 nucleotides, with lengths not divisible by 3 or with internal stop codons, leaving 2142 sequences of H1 and 49543 sequences of H3. We then used cd-hit program [63] to discard identical sequences from the dataset, leaving 1557 sequences of H1 and 20191 sequences of H3. Finally, we randomly subsampled 1557 sequences of H3 to make the H1 and H3 datasets equal in size. Protein sequences were obtained from DNA sequences with BioPerl scripts. All subsequent procedures were conducted as for the HIV-1 dataset. We obtained experimentally derived amino acid propensities for A/WSN/1933 (H1N1) and A/Perth/16/2009 (H3N2) strains from [27]. The initial length of these samples

was 565 and 566 amino acids, respectively. The initial protein alignment included 580 positions, and 550 positions containing on average 3.8 different amino acids per site remained after filtering of positions with more than 1% of gaps or ambiguous characters (with 2.9 and 2.3 amino acids per site in subtypes H1N1 and H3N2, respectively). Among those, DMS results for both focal strains were available for 537 sites.

## Simulated evolution

We simulated evolution of the protein across the phylogeny of two HIV-1 subtypes, B and C. The phylogeny was constructed as above, but only sequences from B and C subtypes were used. The "sparse" tree was constructed by removing a fraction of terminal branches from the real tree, and the "dense" tree, by adding, to the end of each terminal node of the initial tree, two descendant branches ending with new terminal nodes, with lengths uniformly picked from the range (0;0.2], which roughly matches the terminal branch lengths of the reconstructed *env* trees. We constructed these trees using BioPerl functions (https://bioperl.org/).

The second tree that we used for simulations is a tree of five mitochondrially encoded proteins from [20] with topology based on taxonomy and branch lengths corresponding to numbers of amino acid substitutions per site.

Simulations were performed with the Simulator of Evolution with Landscape Variation (SELVa) program [30]. SELVa allows to specify the branches of the phylogenetic tree where a fitness shift occurs, and the vectors of amino acid fitness before and after the shift. It simulates substitutions between amino acids according to the model described in [31]. The mutation rates are assumed to be uniform, so the differences in substitution rates are solely determined by differences in fitness.

In SELVa, a fitness landscape is represented by a vector of scaled additive fitness values for each amino acid (or any character in the sequence alphabet). We simulated evolution of 500 sites along the phylogenetic tree of HIV-1 with the scaled amino acid fitness vector (X, 1, 1, 1, 1, 1, 1, 1, 1, 1, 1, 1, 1, 1, 1, 1, 1, 1, 1, 1) (where each number corresponds to the fitness of one of the 20 amino acids, see Section "Method validation with simulations and data" of the Results) for each integer $X$ between 3 and 10; in other words, at each instance, one of the amino acids had fitness X, while the remaining 19 had fitness 1. The identity of the amino acid with scaled fitness $X$ differed between subtypes B and C; the corresponding fitness shift was assumed to have occurred in the last common ancestor of subtype C.

For each X, we calculated two values: the true positive rate (TPR)–the proportion of all amino acids across all sites, among those that have changed fitness in the course of evolution and were found at the site, that were correctly identified as positives; and the false positive rate (FPR)–the proportion of all amino acids across all sites, among those that have not changed fitness and were found at the site, that were incorrectly identified as positives. These values were calculated for 1000 randomly sampled amino acids for 100 times to obtain the mean and the 90% confidence interval.

For simulations described in Table D in S2 File, TPR and FPR were calculated only for the testable amino acids across all sites of the protein. Each amino acid was considered as a single observation independently of the type of fitness vector of a site.

## Comparison with DMS data

To test whether our results match the results of the DMS experiments, for each amino acid at each site, we measured the ratio $r$ between the DMS preference for this amino acid in the strain for which it is phylogenetically more proximal ($\pi_x$) and in the strain for which it is more distal ($\pi_y$). The proximity to one or the other strain was determined by comparison of the

corresponding p-values, independent of their significance, so that all amino acids could be thus classified. We then compared the mean $r$ for variable fitness and other amino acids with Wilcoxon's rank sum test.

## Identifying sites under positive selection

Sites of gp160 were tested for positive selection using the codeml program of the PAML package with site model. The presence of positive selection was tested by comparison of models M1a and M2a with the likelihood ratio test. Probabilities that positive selection acts at each site were calculated by BEB (Bayes Empirical Bayes) method as implemented in codeml. We separately performed the analysis for subtype A subtree, subtype B subtree and A+B tree. The trees for these analyses were built as described above, but only sequences from subtypes A and/or B were used.

## Studying association of variable fitness amino acids with differences of antigenic properties between HIV1 subtypes

We used IC50 values for 19 subtype A strains, 31 subtype B strains, and 28 subtype C strains for bnAbs VRC01, NIH45-46, b12, PG9, 2G12, 2F5, and 4E10 from [36]; for 8 subtype A strains, 13 subtype B strains, and 12 subtype C strains for bnAbs PGT121 and PGT128 also from [36]; and for 31 subtype A strains, 48 subtype B strains, and 59 subtype C strains for bnAbs VRC01 and PG9 from [35]. For each bnAb, we performed a two-sided Mann-Whitney test to understand whether its neutralizing capacity differs between subtypes A and B or subtypes B and C.

Then we mapped sites with the variable fitness amino acids onto the overlay of multiple gp120/bnAb complexes: with VRC01 (PDB id 3ngb), PG9 (PDB id 3u4e), b12 (PDB id 2ny7), NIH45-46 (PDB id 4jkp), 2G12 (PDB id 6e5p), PGT121 (PDB id 5c7k), and PGT128 (PDB id 5cez). Epitopic sites were defined according to [37].

To show all considered antibodies in connection with gp120 on one picture (Fig 6A), we combined structures 6e5p, 4jkp, 3ngb, 2nz7, 5cez and 5c7k.

Throughout the paper, statistical analyses (e.g. Mann-Whitney test, Fisher's exact test) were performed with functions of basic R language [64].

## Supporting information

**S1 File. Supplementary figures A–J.** Fig A in S1 File. Phylogenetic tree of B and C HIV-1 subtypes used in simulation analysis. Green clade is subtype B; blue clade is subtype C; black dots are the two strains to which we applied our test in simulation analysis; red dot is a point of fitness shift. Fig B in S1 File. False Positive Rate of our approach for phylogenetic trees with ~1650, 3300 and 6600 species in simulations for different scaled fitness values of the preferred variant X; black line, median; grey area, 90% confidence interval from random sampling of 1000 amino acids for 100 times. Upper row: all amino acids are considered, bottom row: only testable amino acids are considered. Fig C in S1 File. Performance of the d-test for a phylogenetic tree of subtypes B and C with all branches squeezed or stretched 2 or 4 times, in simulations for different scaled fitness values of the preferred variant. Red line, median True Positive Rate (TPR); blue line, median False Positive Rate (FPR). 90% confidence bands were obtained by random subsampling of 1000 amino acids in 100 trials. Upper row: all amino acids are considered, bottom row–only testable amino acids are considered. Fig D in S1 File. Various conditions for simulations described in Tables C, D in S2 File. Blue dots—focal nodes, red dots—alternative points of shift in fitness vectors. Fig E in S1 File. Schematic representation of cases that can decrease performance of the d-test. Red dots, focal nodes; blue cross, point of fitness

shift. **A**, ideal case; **B**, both focal points have the same fitness vector, but fitness shift occurs closer to one of them, leading to false positives; **C**, there is a long branch between the fitness shift and one of the focal points, leading to false negatives; **D**, distance between the two focal points is less than the mean distance between one of them and species with the same fitness landscape, leading to false negatives. Fig F in S1 File. Performance of the d-test for a phylogenetic tree of Metazoa and Fungi (Fig D (B)) or its subtree for Vertebrates (Fig D(C)), in simulations for different scaled fitness values of the preferred variant. Red line, median True Positive Rate (TPR); blue line, median False Positive Rate (FPR). 90% confidence bands were obtained by random subsampling of 1000 amino acids in 100 trials. Upper row: all amino acids are considered, bottom row–only testable amino acids are considered. Fig G in S1 File. Performance of GEMME for a phylogenetic tree of subtypes B and C, in simulations for different scaled fitness values of the preferred variant. An amino acid in a site was considered significant if it has positive score in one subtype and negative score in the other subtype. Red line, median True Positive Rate (TPR); blue line, median False Positive Rate (FPR). 90% confidence bands were obtained by random subsampling of 1000 amino acids in 100 trials. Fig H in S1 File. Performance of prevalence-based test for a phylogenetic tree of subtypes B and C, in simulations for different scaled fitness values of the preferred variant. Red line, median True Positive Rate (TPR); blue line, median False Positive Rate (FPR). 90% confidence bands were obtained by random subsampling of 1000 amino acids in 100 trials. Fig I in S1 File. Ratio of experimentally measured fitness is subtype for which amino acid is proximal and subtype for which amino acid is distal is higher for amino acids that are significant in d-test than amino acids that are significant in prevalence-based test. Significance value of two-sided Wilcoxon rank sum test is shown. Fig J in S1 File. Q-value for different significance thresholds obtained as min $(\alpha^*P)/V$, where $\alpha$ is the significance threshold, P is the number of testable amino acids at this $\alpha$, and V is the number of significant tests at this $\alpha$, in testing of the hypothesis that an amino acid is proximal or distal for one focal node. Black, gp160; grey, HA; solid, hypothesis of proximity; dashed, hypothesis of distality. Dotted lines correspond to the significance threshold used in the main analysis for gp160 (black) and HA (grey).
(PDF)

**S2 File. Supplementary tables A-F.** Table A in S2 File. Power of d-test in simulations with different selective constraints for three sample sizes (median values of TPR and FPR for 100 random subsamples of 1000 amino acids). Table B in S2 File. Power of d-test in simulations with different selective constraints for four evolutionary rates (median values of TPR and FPR for 100 random subsamples of 1000 amino acids). Table C in S2 File. Performance of the d-test under different simulation conditions. Phylogenetic tree with points of fitness shift is shown in Fig D (A) in S1 File. *—cases when fitness shift occurred in a phylogenetic neighborhood of one focal node, but both focal nodes had the same fitness vector. Fitness of amino acids with changing preferences is shown in red. Sensitivity and specificity that maximized Yoden's coefficient (TPR-FPR) and therefore may serve as the "optimal" performance are shown in "max (Yoden's coeff)" column. The performance was calculated across all sites of the protein using only testable amino acids. Table D in S2 File. Power of d-test in simulations with different selective constraints for mitochondrial tree (Fig D (B) in S1 File) and its subtree (Fig D (C) in S1 File) (median values of TPR and FPR for 100 random subsamples of 1000 amino acids). Table E in S2 File. Some of variable fitness amino acids are located in sites with different antigenicity in viral subtypes. Table F in S2 File. Consensus amino acids in a pair of viral subtypes in sites with variable fitness amino acids that reside in epitopes of antibodies with different binding capacity in two subtypes.
(DOCX)

**S3 File. Supplementary text: Robustness of the results of the d-test for different focal nodes and for random subsets.**
(PDF)

**S1 Table. Differences in fitness constraints between subtypes A, B and C of HIV-1.**
(XLSX)

## Acknowledgments

We thank Juhye Lee, Hugh Haddox and Jesse Bloom for sharing their DMS data pre-publication and for valuable comments. We also thank Alexey Neverov for helpful discussions.

## Author Contributions

**Conceptualization:** Galya V. Klink, Georgii A. Bazykin.

**Formal analysis:** Galya V. Klink, Olga V. Kalinina.

**Funding acquisition:** Galya V. Klink.

**Supervision:** Georgii A. Bazykin.

**Visualization:** Galya V. Klink, Olga V. Kalinina.

**Writing – original draft:** Galya V. Klink.

**Writing – review & editing:** Galya V. Klink, Georgii A. Bazykin.

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
