## [Decision Letter · Decision Letter 0]

24 Aug 2021

Dear Dr. Klink,

Thank you very much for submitting your manuscript "Phylogenetic inference of changes in amino acid propensities with single-position resolution" for consideration at PLOS Computational Biology.

As with all papers reviewed by the journal, your manuscript was reviewed by members of the editorial board and by several independent reviewers. In light of the reviews (below this email), we would like to invite the resubmission of a significantly-revised version that takes into account the reviewers' comments.

We cannot make any decision about publication until we have seen the revised manuscript and your response to the reviewers' comments. Your revised manuscript is also likely to be sent to reviewers for further evaluation.

Sincerely,

Marco Punta

Associate Editor

PLOS Computational Biology

Arne Elofsson

Deputy Editor

PLOS Computational Biology

Reviewer's Responses to Questions

**Comments to the Authors:**

Reviewer #1: The article presents a method to estimate fitness changes for amino-acids in fixed positions between subtypes/clades. It is presented as a general method to be applied to protein families but tested exclusively on viral proteins.

The method is of interest but I found that the presentation is not straightforward, the testing is limited, the method is not described clearly, the comparison with tools that estimate phenotypic effects is not performed.

Some remarks:

- Protein families are not presented precisely: sequence length, number of sequences in the set, sequence conservation level, conservation levels and number of sequences within the clades containing focal points used in the article….

- As mentioned above, the approach is presented as a general approach, applicable to arbitrary protein families but tested on viral proteins only. This should be discussed and if the authors think that the method can be applied to other protein families, they should say it and provide testing. The problem might be to find available experimental data allowing for a comparative evaluation of the fitness within different clades. If this is the case, then a comparison with computationally generated data is important. See below.

- Overall the writing is unclear with notions that are not completely standard. For instance, what is a “segment” of a phylogenetic tree (pp4)? Is it a branch point? Maybe use more common notions.

Other examples are:

pp 28. The phrase “by adding, to each terminal node of the initial tree, two new branches”… what does this mean? How general this way to do is? Can you envisage some other operation?

pp29. The vector defined as {X,1,….,1,1,1} is not a way to write it! The 20 “1”s correspond to what? I counted the number of “1”s btw and some wording could be added. How does this vector evolve? A formal description of your method is needed. It would make the whole message much clearer.

- There is no comparison with the matrix reconstruction of DMS experiments proposed by other groups. The method identifies a subset of significant positions in a sequence and correlates them with DMS measures. Today, we have methods reconstructing matrices from DMS experiments that perform very well. See, for instance, besides Hopf et al. 2017 and Riesselman et al. 2018 also

Laine, E., Karami, Y., & Carbone, A. (2019). GEMME: a simple and fast global epistatic model predicting mutational effects. Molecular biology and evolution, 36(11), 2604-2619.

Munro, D., & Singh, M. (2020). DeMaSk: a deep mutational scanning substitution matrix and its use for variant impact prediction. Bioinformatics, 36(22-23), 5322-5329.

The model introduced by the authors should be compared to the performance of these methods. They are based on different ideas and it is important that they discuss them. Especially the GEMME method, based on trees, could be considered and discussed on the methodological site. The authors seem to ignore these developments. The model introduced in this work could gain in impact if the author can show its interest in others predictions.

- As discussed, in this article there is no reference to the genetic context provided by the entire sequence that in principle could be relevant for predicting the phenotype as demonstrated in the works above. This point should be discussed on the light of precise computational comparisons.

- A careful combined analysis of parameters like the size of the clades and their variability in sequence identity/similarity should be realized.

- The quality of the figures is very poor. Small points with indistinguishable colors.

Reviewer #2: Review is uploaded as an attached PDF.

Reviewer #3: In this manuscript, the authors proposed a statistical d-test to identify changing propensity of amino acid states across phylogeny, by measuring the distance of certain types of substitution to focal taxa in the tree. Proximation to one taxon and distality to another indicate increase of propensity, and hence fitness, of the resulting amino acid to the former clade. The authors use simulation to show its sensitivity and specificity, then show that the detected substitutions are consistent with fitness change measured by deep mutational scanning, while not all of them are under positive selection. The authors then describe the relation between d-test detected variable fitness amino acids with immunogenicity in HIV and Influenza viruses.

Since the test depend heavily on the amino acid states appearing at each site, although the authors have made significant efforts in analyzing the statistical properties of the test, I am still concerned with how stable the test result is, i.e. for the same set of data, how variable the detection power and the accuracy is:

(1) To conduct d-test, two or more focal nodes needs to be selected for measuring d. With different focal nodes assigned for the same two clades, would d-test detect the same set of variable fitness amino acids?

(2) Ten independently reconstructed trees were used in one d-test, but with the same set of taxa. If the propensity change is true for a clade, then variation of taxa included in a clade (not necessarily changing the number of taxa) should not affect the detection results. Is this true?

To confirms the superiority of the d-test, there should be a baseline method for control. For example, if one simply compares the observed frequency of amino acid states in each clade and test the significance by some randomized null distribution, how would the performance be worse than the d-test?

Around Line 371-404, it is argued that the variation of fitness is associated with difference in antigenic binding. However, there is no statistical significance. Specifically, I cannot understand why the binding could be affected if the most prevalent amino acid does not change.

Table S4: It seems there are two rows with VRC01 and PG9. I could not find the explanation for their difference. Besides, is there multiple test correction for p-values in this table?

**Have the authors made all data and (if applicable) computational code underlying the findings in their manuscript fully available?**

Reviewer #1: Yes

Reviewer #2: Yes

Reviewer #3: Yes

PLOS authors have the option to publish the peer review history of their article (what does this mean?). If published, this will include your full peer review and any attached files.

Reviewer #1: No

Reviewer #2: No

Reviewer #3: No
---

## [Decision Letter · Decision Letter 1]

28 Jan 2022

Dear Dr. Klink,

We are pleased to inform you that your manuscript 'Phylogenetic inference of changes in amino acid propensities with single-position resolution' has been provisionally accepted for publication in PLOS Computational Biology.

Best regards,

Arne Elofsson

Deputy Editor

PLOS Computational Biology

Arne Elofsson

Deputy Editor

PLOS Computational Biology

We trust you to make the minimal corrections (formula, number of significant numbers etc) during the proofing of the manuscript.

Reviewer's Responses to Questions

**Comments to the Authors:**

Reviewer #1: The authors have addressed all my points and responded satisfactorily.

Reviewer #2: I raised only minor issues when first reviewing this ms, and the authors have adequately addressed them through their extensive textual additions on pages 10-11, in the Methods and in the Discussion clarifying more details about the extensive simulations undertaken. I have no more important concerns to raise and look forward to seeing this ms in print.

One last nitpick – where you use an algebraic symbol in the axis labels of your figures (X, r, π), perhaps add text in the label explaining what it means (rather than just in the caption). E.g. in Figure 3, you could label X as “X (fitness value)” or similar.

Reviewer #3: The questions has largely been properly answered. My additional suggestion is, to edit the statistical conclusions to be more congruent.

For example, there should not be four decimal digits for a P-value ("We obtained significant, albeit rather weak, correlations of fitness differences obtained by our method with those of GEMME (p < 2.2e-16, rho = 0.24) and DeMask (p = 7.048e-16, rho = 0.11) scores"), and P-value could be a precise value rather than "< 2.2e-16".

Also, in Table S4 and so on, a multiple test correction of P-values may change the results quantitatively.

Page 10: "Specifically, the specificity remains high when more several amino acids change their fitness". Typo for "more several"?

**Have the authors made all data and (if applicable) computational code underlying the findings in their manuscript fully available?**

Reviewer #1: Yes

Reviewer #2: Yes

Reviewer #3: Yes

PLOS authors have the option to publish the peer review history of their article (what does this mean?). If published, this will include your full peer review and any attached files.

Reviewer #1: No

Reviewer #2: No

Reviewer #3: No

---

## [Editor Report · Acceptance letter]

14 Feb 2022

PCOMPBIOL-D-21-01056R1 

Phylogenetic inference of changes in amino acid propensities with single-position resolution

Dear Dr Klink,

I am pleased to inform you that your manuscript has been formally accepted for publication in PLOS Computational Biology. Your manuscript is now with our production department and you will be notified of the publication date in due course.

With kind regards,

Agnes Pap
